# A Novel Biplex *Onchocerca volvulus* Rapid Diagnostic Test Evaluated Among 3- to 9-Year-Old Children in Maridi, South Sudan

**DOI:** 10.3390/diagnostics15050563

**Published:** 2025-02-26

**Authors:** Amber Hadermann, Stephen Raimon Jada, Charlotte Lubbers, Luís-Jorge Amaral, Marco Biamonte, Dziedzom Komi de Souza, Yak Yak Bol, Joseph Nelson Siewe Fodjo, Robert Colebunders

**Affiliations:** 1Global Health Institute, University of Antwerp, 2610 Antwerp, Belgium; amber.hadermann@uantwerpen.be (A.H.); charlottelubbers@hotmail.nl (C.L.); luis-jorge.telesdemenesesdoamaral@uantwerpen.be (L.-J.A.); josephnelson.siewefodjo@uantwerpen.be (J.N.S.F.); 2Amref Health Africa, Juba P.O. Box 410, South Sudan; stephen.jada@amref.org; 3Drugs & Diagnostics for Tropical Diseases, San Diego, CA 92121, USA; marco.biamonte@ddtd.org; 4Noguchi Memorial Institute for Medical Research, College of Health Sciences, University of Ghana, Accra P.O. Box LG 581, Ghana; ddesouza@noguchi.ug.edu.gh; 5Neglected Tropical Diseases Programme, Ministry of Health, Juba P.O. Box 88, South Sudan; yakdit16@gmail.com; 6Department of Tropical Biology, Liverpool School of Tropical Medicine, Liverpool L3 5QA, UK

**Keywords:** onchocerciasis, *Onchocerca volvulus*, antibodies, transmission, testing, seroprevalence, children, feasibility

## Abstract

**Background**: Point-of-care diagnostic tests are essential for confirming *Onchocerca volvulus* transmission in remote, resource-limited, onchocerciasis-endemic communities. In Maridi, South Sudan, we field-tested a novel “biplex A” rapid diagnostic test (RDT) developed by Drugs & Diagnostics for Tropical Diseases (DDTD), San Diego, California. **Methods**: In February 2023, children aged 3–9 years were recruited from study sites at different distances from the Maridi Dam, a known blackfly breeding site. *O. volvulus* antibodies were detected using the DDTD biplex A RDT, which detects antibodies to Ov16 and OvOC3261 at test line 1 and to Ov33.3 and OvOC10469 at test line 2, along with the commercially available Ov16 SD Bioline RDT. Both tests were performed on whole blood obtained via finger prick. The feasibility and acceptability of the DDTD biplex A RDT were assessed, and its results were compared with those of the Ov16 SD Bioline RDT. **Results**: A total of 239 children participated in the study. The anti-Ov16 seroprevalence detected by the Ov16 SD Bioline RDT was 30.2% (72/239), with the highest prevalence observed in children living closest to the Maridi Dam (*p* < 0.001). Testing with the DDTD biplex A RDT was determined to be feasible, acceptable, and easy to use in a field setting. The DDTD biplex A RDT test line 1 (anti-Ov16 and anti-OvOC3261) was positive in 35.1% (84/239) of children, while test line 2 (anti-Ov33.3 and anti-OvOC10469) was positive in 18.4% (44/239). Both lines were simultaneously visible in 15.5% (37/239). **Conclusions**: The DDTD biplex A RDT prototype was user-friendly and practical for field deployment. However, additional research is needed to evaluate its performance relative to the commercially available Ov16 SD Bioline RDT. The high anti-Ov16 seroprevalence that was observed underscores the ongoing *O. volvulus* transmission near the Maridi Dam. Strengthening the onchocerciasis elimination program in Maridi should be prioritized as a critical public health intervention.

## 1. Introduction

Onchocerciasis or river blindness, caused by infection with *Onchocerca volvulus,* is a neglected tropical disease that is associated with significant morbidity and disability, including skin and eye lesions [1] and onchocerciasis-associated epilepsy [2]. The life cycle of *O. volvulus* involves both a human host and a blackfly (*Simulium* species) vector. Infective third-stage larvae (L3) are transmitted to humans through the bite of an infected blackfly. The larvae mature into adult worms in subcutaneous nodules, where they can live for years and produce the microfilariae—the primary agents of disease pathology [1].

The antigens of *O. volvulus* include a large variety of structural, metabolic, and excretory–secretory proteins that play crucial roles in the parasite’s survival, immune evasion, and host–pathogen interactions [3]. These antigens are also key targets for diagnostics, vaccines, and drug development. A total of 7774 *O. volvulus* proteins were identified across all of the stages [4]. The OV16 antigens have been shown to elicit strong IgA responses and are, therefore, used as the main antigen in serodiagnostic tests. Other antigens, such as the OvOC3261, Ov33.3, and OvOC10469 antigens, are currently under evaluation [5].

The World Health Organization (WHO) aims to interrupt the transmission of *Onchocerca volvulus* in at least 12 countries by 2030 and to have all endemic countries in the elimination phase by then [6]. In this context, Ov16 antibody seroprevalence surveys using Ov16 ELISA techniques have been proposed to assess the impact of interventions in endemic areas [6]. Ov16 is a diagnostic antigen derived from *O. volvulus* that elicits a strong IgG4 response in infected individuals, making it a key target for serological tests [7]. Several enzyme-linked immunosorbent assay (ELISA) protocols have been developed, including the Onchocerciasis Elimination Program for the Americas (OEPA) ELISA, the Centers for Disease Control and Prevention (CDC) ELISA, and the SD Bioline ELISA. However, performing these tests requires transporting samples to a specialized laboratory, which depends on a well-maintained cold chain, specialized equipment, and trained personnel. These logistical hurdles led to the increased adoption of the Ov16-antibody (anti-Ov16) rapid diagnostic tests (RDTs) for field testing. RDTs are a more practical, point-of-care solution in remote villages, although they may not match the sensitivity and specificity of a laboratory-based ELISA [8].

Comparing the performances of Ov16 SD Bioline RDT (Abbot Diagnostics Inc, Republic of Korea) with laboratory-based anti-Ov16 ELISA has yielded different outcomes in different populations. Among persons with onchocerciasis-associated epilepsy in the Democratic Republic of Congo (DRC), the anti-Ov16 ELISA performed on the participants’ stored sera was more sensitive (83.0%) than the Ov16 SD Bioline RDT (74.8%, *p* = 0.12) performed on freshly collected blood, although less sensitive (respectively, 84.8% versus 98.6%, *p* ≤ 0.01) [8]. These sensitivities were determined by comparing each test with the combined results of the other two tests (i.e., the skin snip and whichever anti-Ov16 test was not being evaluated). In contrast, a 99.2% concordance was found between ELISA on dried blood spots (DBSs) and Ov16 SD Bioline RDT results (performed on fresh blood) in a study among the general population in rural Cameroon [9].

The WHO currently recommends the Ov16 SD Bioline RDT on DBS for monitoring onchocerciasis elimination [10]. However, this test does not fully meet the requirements needed for late-stage control or elimination programs. For onchocerciasis elimination mapping (OEM), diagnostics with high specificity are needed to minimize the likelihood of false-positive results influencing treatment decisions. Conversely, in places where stopping mass drug administration (MDA) is under consideration, diagnostics must also be sensitive enough to detect even the weakest positive cases, ensuring that regions requiring further treatment are not overlooked. The presence of Ov16 antibodies in young children is a measure of infection and, therefore, suggests ongoing *O. volvulus* transmission.

In 2021, the WHO published two target product profiles (TPPs) [11] outlining the key specifications for new diagnostic tests to address two immediate needs: (a) mapping onchocerciasis in areas of low prevalence and determining when to initiate MDA (“Mapping TPP”) and (b) determining when to stop MDA programs (“Stopping TPP”). The preferred format is a field-deployable RDT, ideally a lateral flow assay (LFA), as WHO prioritizes this format for its suitability in low-resource settings where *O. volvulus* is often endemic. For onchocerciasis elimination mapping, the test needs to have ≥60% sensitivity and ≥99.8% specificity, whereas supporting stop-MDA decisions requires a more accurate test with ≥89% sensitivity and ≥99.8% specificity [10]. These stringent sensitivity and specificity requirements ensure the ability to detect infection prevalence as low as 1–2% with high statistical confidence while avoiding false positives (e.g., due to cross-reactivity with other filarial parasites).

Based on a WHO initiative to improve *Onchocerca volvulus* RDT performance in terms of sensitivity and specificity, Drugs & Diagnostics for Tropical Diseases (DDTD), San Diego, California, has created a new serological RDT for onchocerciasis [12]. A first prototype (version A) was formatted as a biplex LFA incorporating four antigens across two distinct test lines. Line 1 detects the presence of human IgG4 antibodies specific for Ov16 and OvOC3261 antigens. Line 2 detects IgG4 antibodies specific for Ov33.3 [13] and OvOC10469 [4]. A test is considered positive only if both lines are visible. This strict interpretation criterion is expected to reduce sensitivity but enhance overall specificity, aligning with the primary objective of improving the test accuracy.

The IgG4 subclass was chosen because it is the primary immunoglobulin isotype associated with chronic helminth infections, such as *O. volvulus* [7]. Although IgG4 represents only a minor fraction of total IgG, it is believed to be produced in response to prolonged antigenic stimulation and correlates closely with infection intensity. In a non-human primate model, IgG4 responses were closely linked to microfiladermia and declined in parallel with parasite clearance, whereas other antibody subclasses (e.g., IgM, IgG2, and IgG3) were less informative [14].

It is important to note that the different antigens involved (Ov16, Ov33.3, OvOC3261, and OvOC10469) are associated with different parasite life stages. Ov16 is expressed by infective L3-stage *O. volvulus* larvae; therefore, humans can develop antibodies to Ov16 upon exposure to an infective *Simulium* bite—*Simulium* being the blackfly vector of *O. volvulus*—even if the infective L3 larvae fail to mature into adults [9]. In contrast, Ov33.3 is expressed by adult *O. volvulus* worms, and OvOC3261 and OvOC10469 are expressed by *O. volvulus* microfilariae, meaning a serological response to these antigens occurs only after adult worms have developed and reproduced [10]. Furthermore, OvOC3261-specific IgG4 antibodies appear to wane faster after treatment compared to Ov16-specific antibodies [15].

While Ov16 is a well-established biomarker in the field of onchocerciasis, to the best of our knowledge, OvOC3261, OvOC10469, and Ov33.3 have not yet been evaluated in a field study. It was, therefore, of interest to evaluate this new DDTD biplex A RDT in the field alongside the Ov16 SD Bioline RDT during a serosurvey among children in Maridi County, an onchocerciasis-endemic area in South Sudan. This first field evaluation of the DDTD biplex A RDT compares its sensitivity to that of a monoplex anti-Ov16 test. Additionally, the study sought to gather user feedback regarding the ease of use and feasibility of implementing this novel test in field conditions. The study was not designed to assess the specificity of the DDTD biplex A RDT, as specificity is evaluated in non-endemic areas.

## 2. Materials and Methods

### 2.1. Study Setting

South Sudan has multiple onchocerciasis-endemic hotspots, including Maridi County in Western Equatoria State. Maridi County, with a population of over 115,000 individuals [16], is intersected by the Maridi River, where a dam was constructed in the 1950s. The spillway of this dam has been identified as the sole breeding site for blackflies in the area [17], driving onchocerciasis transmission in nearby villages (Figure 1).

To address the high transmission, the onchocerciasis elimination program in Maridi employs two key strategies: (1) biannual community-directed treatment with ivermectin (CDTi) and (2) a community-based “slash and clear” vector control method, which reduces the blackfly population by removing vegetation from the dam’s surface [17].

This study was conducted in February 2023 across five sites. Kazana 1, Kazana 2, and Hai-Matara, located near the dam, represent the high-transmission zone, while Hai-Tarawa and Hai-Gabat, situated farther from the dam, constitute the low-transmission zone.

### 2.2. Study Procedures

In February 2023, a cross-sectional study was conducted in selected villages of Maridi. Several weeks prior to the survey, the research team engaged with local community leaders to inform them about the study and secure their approval and collaboration. One day before participant recruitment, community mobilizers informed villagers about the study and the designated testing site in each village where eligible children aged 3 to 9 years would be tested. These sites included schools, churches, and other central gathering places.

On the day of the study in each village, mobilizers used handheld megaphones to announce the study, while healthcare workers (HCWs) conducted house-to-house visits to encourage participation. Children whose adult caretakers provided informed consent were enrolled in the study. Additionally, verbal assent was obtained from children aged 7 to 9 years.

A brief questionnaire was administered to both the caretaker and the child to gather relevant information, including the child’s age, sex, village of residence, previous ivermectin intake, and any history of itching, skin lesions, or epilepsy. Subsequently, all participating children underwent a finger-prick procedure using a 2 mm retractable lancet (Ergo Lance).

A few drops of blood were obtained from the participant’s finger and used for (i) an Ov16 SD Bioline RDT (Abbott, IL, USA); and (ii) a DDTD biplex A RDT. Both rapid tests were performed according to the manufacturers’ instructions, which stipulated a 30 min waiting period before reading the final test results. For each participant, photos of both RDT types were taken at 20 min, 30 min, and 60 min after testing. Paper tape was used to indicate the initial testing time and the read-out times for monitoring purposes. Timings were carefully tracked using a smartphone, though any other type of chronograph could be used if photographing the RDTs was not necessary. All data were collected electronically using the REDCap online platform [18] on electronic tablets.

Five HCWs from diverse medical backgrounds and training levels (including a nurse, two clinical officers, and two midwives) were trained in a two-hour session to perform the DDTD biplex A RDT. The feasibility and acceptability of the DDTD biplex A RDT were assessed by administering a standardized questionnaire (Appendix A) to these HCWs at the end of the serosurvey.

### 2.3. Data Analysis

Study data were reviewed and uploaded into the REDCap secure online platform daily. The final datasets were exported from REDCap and cleaned in Excel spreadsheets, after which they were transferred to the software R version 4.2.2 for analysis. The working assumption was that a DDTD Biplex A RDT was considered positive only when both test lines (T1 and 2) were visible. Other assumptions were added to investigate the individual contribution of each test line, including (1) “T1 or T2”: a test result was considered positive if at least one of the lines (T1 or T2) was positive; (2) “All T1”: a test result was considered positive if T1 was positive, regardless of T2; (3) “All T2”: a test result was considered positive if T2 was positive, regardless of T1; (4) “Only T1”: a test result was considered positive if T1 was positive and T2 was negative; and (5) “Only T2”: A test result was considered positive if T2 was positive and T1 was negative. Continuous variables were summarized as medians with interquartile range (IQR), while categorical variables were expressed as percentages.

## 3. Results

### 3.1. Description of Study Participants

A total of 239 children aged 3–9 years participated in the study. The median age was 6 years (IQR: 4–8), and 46% of participants were males. Ninety-two (37.1%) children exhibited signs of dermatitis, as evidenced by itching and/or visible skin lesions. Furthermore, four (1.7%) children were identified as having epilepsy based on their medical history.

### 3.2. Comparison Between the Prototype DDTD Biplex A RDT and the Ov16 SD Bioline RDT Results

The *O. volvulus* seroprevalence among participants, as determined by Ov16 SD Bioline RDT and DDTD biplex A RDT, is shown in Table 1. The seroprevalence detected by the Ov16 SD Bioline RDT was 30.1% (72/239), whereas the DDTD biplex A RDT (requiring both the T1 and T2 lines to be visible) yielded a significantly lower seroprevalence of 15.5% (37/239; *p* < 0.001) (Table 1).

Both tests indicated that seroprevalence was highest in Kazana 1 and Kazana 2—the sites bordering the breeding site at the Maridi Dam—and was statistically significantly higher than in the other three sites (chi-squared *p*-value(*p*) < 0.001 for both tests). With the Ov16 SD Bioline RDT, significantly more children with dermatitis tested seropositive compared to those without dermatitis (*p* = 0.002). Although the DDTD biplex A RDT did not show a statistically significant difference in seroprevalence between children with and without dermatitis (*p* = 0.14), a higher prevalence was still observed among children with dermatitis (20.7% versus 12.6% without dermatitis). Both RDT tests showed no significant difference in seroprevalence between children with and without a history of ivermectin intake (Ov16 SD Bioline RDT *p* = 0.33; DDTD biplex A *p* = 0.71) nor between sexes (Ov16 SD Bioline RDT *p* = 0.86; DDTD biplex A RDT *p* = 0.85) and across age groups (Ov16 SD Bioline RDT *p* = 0.72; DDTD biplex A *p* = 0.12). Of the four children with epilepsy, none tested positive on either RDT.

*O. volvulus* seroprevalence per participant characteristics using different DDTD biplex A RDT seropositivity criteria are shown in Appendix A.

### 3.3. DDTD Biplex A RDT: Analysis of Test Lines

With the DDTD biplex A RDT, a total of 91 children tested positive for at least one of the two test lines (“T1 or T2”; Table 2). Amongst those 91 cases, the majority were positive only for line T1 (Only T1: 52%; 47/91), some for both test lines (T1 and T2: 41%; 37/91), and a few only for line T2 (Only T2: 8%; 7/91), making the line distribution 52:41:8.

Both the seroprevalences calculated using “T1 or T2” (38.1%) and “All T1” (35.2%) were found to be significantly higher than the seroprevalence calculated using the manufacturers’ read-out instructions, “T1 and T2”, which was 15.5% (37/239; *p* < 0.001) (Table 2), whereas no significant difference was observed when comparing “T1 and T2” (15.5%) with “All T2” (18.4%; *p* = 0.46) or “Only T1” (19.7%; *p* = 0.28). However, the seroprevalence of “T1 and T2” (15.5%) was significantly higher than “Only T2” (2.9%; *p* < 0.001) (Table 2).

When comparing these seroprevalences to Ov16 SD Bioline RDT (30.1%, 72/239), no significant difference was found with “T1 or T2” (38.1%; *p* = 0.08) nor “All T1” (35.2%; *p* = 0.28). However, Ov16 SD Bioline RDT’s seroprevalence was significantly higher than “All T2” (18.4%; *p* = 0.004), “Only T1” (19.7%; *p* = 0.01), and “Only T2” (2.9%; *p* < 0.001) (Table 2).

Twenty children were found to test positive for at least one line (T1 and/or T2) with DDTD biplex A while testing negative for Ov16 SD Bioline RDT (Table 3). In contrast, only one child tested positive for Ov16 SD Bioline RDT and negative for both of DDTD biplex A’s test lines. However, 41 children tested positive for Ov16 SD Bioline RDT and negative for DDTD biplex A RDT using the “T1 and T2” positivity assumptions suggested by the manufacturers.

### 3.4. Effect of Timing on the Test Results

The timed photographs revealed that 91.2% (83/91) of positive DDTD biplex A RDT tests were positive after 20 min, and the remaining 8.8% (8/91) turned positive after 30 min. In addition, 37% (33/90) of test lines tended to become brighter from 30 to 60 min (e.g., Figure 2).

### 3.5. Feasibility and Acceptability of the Prototype DDTD Biplex A RDT

The prototype DDTD biplex A RDT was found to be feasible, acceptable, and easy to use in the field by all five HCWs who participated in the study. The HCWs mentioned the similarity between the DDTD biplex A RDT and other finger-prick-based RDTs, such as for HIV and malaria, which made them familiar with the required test procedures. A two-hour training session was sufficient for all HCWs to perform the test and correctly complete the participants’ questionnaires. However, there were difficulties in monitoring the timings of several tests simultaneously by the HCWs, especially given the continuous influx of children for whom study consent had to be sought. The most effective way of organizing the HCWs for the RDT research was in teams of two, one to manage participant intake and questionnaire administration and one to perform the test. Additionally, a person designated for test read-outs remained stationed in a central place. Based on observation, calculating read-out time proved difficult, so additional training and supervision to ensure accurate timing of read-outs needs to be organized.

The area surrounding the read-out desk was used as a waiting area for individuals who wanted to be notified of their test results, which facilitated the effective and efficient reporting of results.

## 4. Discussion

The anti-Ov16 seroprevalence was 30.1% (72/239) with the Ov16 SD Bioline RDT, whereas with the DDTD biplex A RDT (both T1 and T2 lines positive), the seroprevalence was statistically significantly lower at 15.5% (37/239; *p* < 0.001). When considering only the DDTD biplex A RDT T1 line, which detects Ov16 and OvOC3261 antibodies, the seroprevalence was 35.1% (84/239), not significantly different from the seroprevalence determined with the Ov16 SD Bioline RDT (*p* = 0.28).

Using the DDTD biplex A RDT, a total of 91 children tested positive for at least one of the two test lines, with a line distribution of 51:41:8, where most children tested positive for T1 only. This distribution contrasts with previously reported distributions by the NIH (9:84:7) and CDC (7:84:9), where most samples tested positive for both test lines [5]. This discrepancy likely reflects differences in study conditions: NIH and CDC evaluations used sera from people with confirmed infection, whereas the present study was a field survey using whole blood obtained by finger-pricking. Moreover, the study involved a younger population that may have been exposed to *O. volvulus* infective bites (Ov16); therefore, they were positive by test line 1 but had insufficient time to develop a patent infection (Ov33.3, OvOC10469) and, therefore, remained negative for test line 2. An age-related increase in anti-Ov16 seroprevalence has previously been observed in another community-based study in the DRC [19]. Both anti-Ov16/anti-OvOC3261 (line T1) and anti-Ov33.3/anti-OvOC10469 (line T2) seropositivity were non-significantly associated with dermatitis, which is mediated by the *O. volvulus* microfilariae. OvOC3261 is thought to be expressed solely or at least primarily by the microfilariae [4]. Further longitudinal studies in children are needed to fully understand the seroconversion dynamics—especially, WHO recommends surveys within this age group (3–9 years old) to monitor the onchocerciasis elimination targets [10].

The finding that anti-Ov16/anti-OvOC3261 seroprevalence (35.1%) was twice as high as the anti-Ov33.3/anti-OvOC10469 seroprevalence (18.4%) is consistent with the differential expression of these antigens across parasite life stages. It is important to note that the DDTD biplex A RDT is considered positive only if both lines are positive (15.5%), and this differs with the higher anti-Ov16 seroprevalence (35.1%). Consequently, if the DDTD biplex A RDT or a similar version is to be used to guide start- or stop-MDA decisions, seroprevalence thresholds may need to be adapted compared to those based solely on anti-Ov16. Refining these thresholds will require further operational research.

The specificity of the DDTD RDT prototype has already been evaluated by the CDC using sera from other infections, and they found the specificity to be >99%. Additional studies, including well-characterized human adult populations from other onchocerciasis-endemic regions, will be needed to assess the sensitivity and specificity of DDTD biplex A RDT relative to a gold standard. Such studies will require skin snip testing and assessment for parasitological co-infections, particularly other filarial parasites capable of eliciting cross-reactive immune responses. Notably, testing for the presence of *Mansonella perstans*, a common co-infection in South Sudan, is crucial to exclude possible cross-reactivity that may influence the specificity of the test [20].

This study had several limitations. For ethical reasons, skin snips were performed to confirm onchocerciasis diagnosis, preventing an assessment of the true sensitivity of the RDTs in microfilariae-positive children. The final RDT set-up required up to three HCWs (two mobile plus one stationary read-out supervisor) and allowed for some door-to-door recruitment by the mobile HCWs. While this approach enhances epidemiological accuracy by better simulating community seroprevalence, it also increases personnel costs if replicated on a larger scale.

## 5. Conclusions

In conclusion, Maridi remains a hotspot for *O. volvulus* transmission despite the implementation of elimination measures. Strengthening these interventions and increasing ivermectin treatment coverage will be crucial to achieving the WHO’s 2030 onchocerciasis elimination goals. A highly sensitive and specific *O. volvulus* RDT suitable for field use would be a valuable tool for planning *O. volvulus* elimination strategies and monitoring progress. Further studies are needed to determine whether the newly optimized DDTD biplex D test could fulfill this role.

## Figures and Tables

**Figure 1 diagnostics-15-00563-f001:**
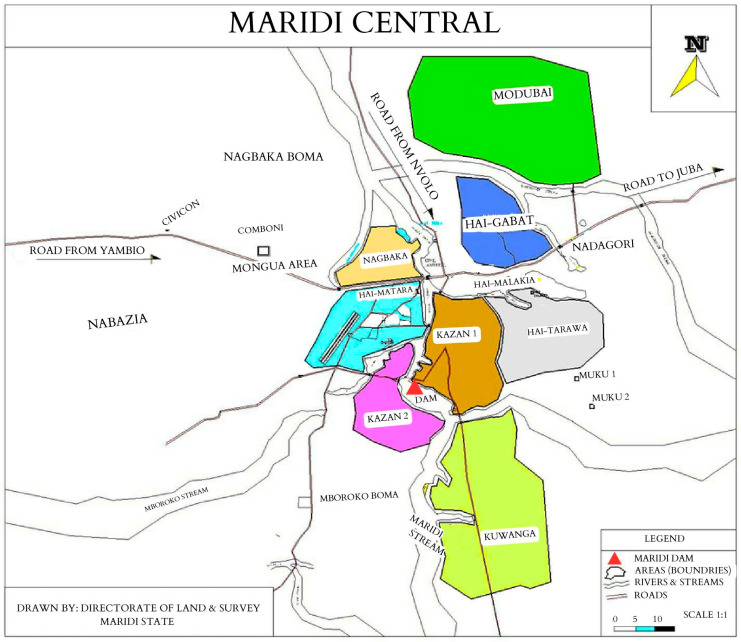
Map of Maridi Central Area, including the five study sites: Kazana 1, Kazana 2, Hai-Matara, Hai-Tarawa, and Hai-Gabat. Kazana 1 and Kazana 2 are located along the Maridi River, with the Maridi Dam (red triangle) linking the two sites. Hai-Gabat is located further north, the most distant study site from the Maridi Dam.

**Figure 2 diagnostics-15-00563-f002:**
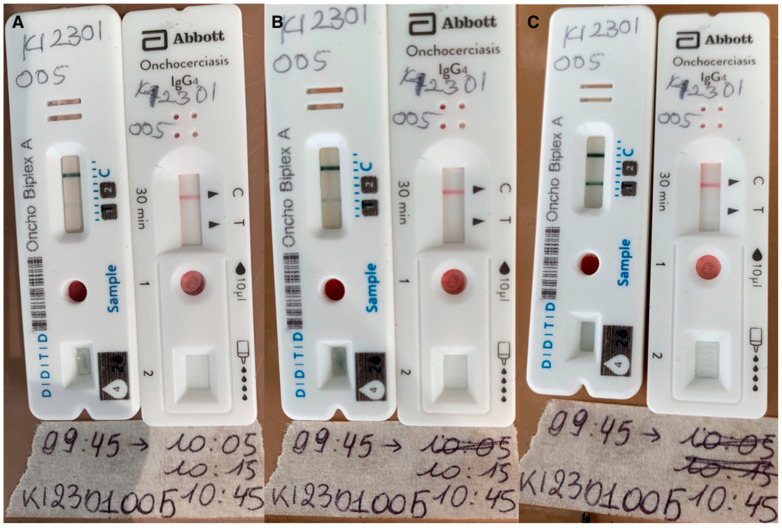
Lateral flow test devices: Positive DDTD biplex A (line 1; left RDT) and negative Ov16 SD Bioline RDT (right) at different time points: (A) 20 min, (B) 30 min, and (C) 60 min. The pieces of paper contain the time of testing, the times the test had to be read, and the code of the study participant.

**Table 1 diagnostics-15-00563-t001:** Ov16 SD Bioline and DDTD biplex A rapid diagnostic test seroprevalence per characteristic, incl. village, sex, ivermectin use, dermatitis, epilepsy, and age group.

Factor	Level	Ov16 SD Bioline RDT Prevalence of Positive Test Result (%; 95%CI)	DDTD Biplex A RDT Prevalence of Positive T1 and T2 Lines Test Result (%; 95%CI)
Village	Hai-Gabat	2/50 (4.0; 0.7–14.9)	0/50 (0; 0–8.9)
	Kazana 1	22/50 (44.0; 30.3–58.7)	11/50 (22.0; 12.0–36.3)
	Kazana 2	20/38 (52.6; 36.0–68.7)	14/38 (36.8; 22.3–54.0)
	Hai-Matara	17/51 (33.3; 21.2–48.0)	9/51 (17.7; 8.9–31.4)
	Hai-Tarawa	11/50 (22.0; 12.0–36.3)	3/50 (6.0; 1.6–17.5)
Sex	Female	40/129 (31.0; 23.3–39.8)	21/129 (16.3; 10.6–24.0)
	Male	32/110 (29.1; 21.0–38.7)	16/110 (14.6; 8.8–22.9)
Ivermectin Intake (2022)	No	46/165 (27.9; 21.3–35.5)	27/165 (16.4; 11.2–23.1)
	Yes	26/74 (35.1; 24.6–47.2)	10/74 (13.5; 7.0–23.9)
Dermatitis	No	34/151 (22.5; 16.3–30.2)	19/151 (12.6; 7.9–19.2)
	Yes	37/87 (42.5; 32.1–53.6)	18/87 (20.7; 13.0–31.0)
Epilepsy	No	72/235 (30.6; 24.9–37.0)	37/235 (15.7; 11.5–21.2)
	Yes	0/4 (0; 0–60.4)	0/4 (0; 0–60.4)
Age group (years)	Age 3–6	46/147 (31.3; 24.0–39.5)	18/147 (12.2; 7.6–18.9)
	Age 7–9	26/92 (28.3; 19.6–38.8)	19/92 (20.7; 13.2–30.6)
Total	72/239 (30.1; 24.5-36.4)		37/239 (15.5; 11.3–20.8)

T1 tests for the presence of Ov16 and OvOC3261 antibodies and T2 for Ov33.3 and OvOC10469 antibodies; CI = confidence interval.

**Table 2 diagnostics-15-00563-t002:** DDTD biplex A rapid diagnostic test seroprevalence using different positivity assumptions.

DDTD Biplex A Positivity Assumption	Seroprevalence (%; 95%CI)
Seroprevalence T1 and T2	37/239 (15.5; 11.3–20.8)
Seroprevalence T1 or T2	91/239 (38.1; 31.9–44.6)
Seroprevalence All T1	84/239 (35.2; 29.2–41.6)
Seroprevalence All T2	44/239 (18.4; 13.8–24.0)
Seroprevalence Only T1	47/239 (19.7; 14.9–25.4)
Seroprevalence Only T2	7/239 (2.9; 1.3–6.2)

T1 tests for the presence of Ov16 and OvOC3261 antibodies and T2 for Ov33.3 and OvOC10469 antibodies. Positivity assumptions include “T1 or T2”, where one of the lines can be positive for the test to be positive; “All T1”, where if T1 was positive, the test was positive; “All T2”, where if T2 was positive, the test was positive; “Only T1”, where a test was positive if only T1 and not T2 was positive; “Only T2”, where the test was positive if only T2 and not T1 was positive. CI = confidence interval.

**Table 3 diagnostics-15-00563-t003:** Cross-table between Ov16 SD Bioline RDT and DDTD biplex A per line.

		DDTD Biplex A RDT				
		No line	Only T1	Only T2	T1 and T2	**TOTAL**
**Ov16 SD Bioline** **RDT**	Negative	147	7	7	6	167
	Positive	1	40	0	31	72
	**TOTAL**	148	47	7	37	

T1 tests for the presence of Ov16 and OvOC3261 antibodies and T2 for Ov33.3 and OvOC10469 antibodies.

## Data Availability

The original contributions presented in this study are available upon request.

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
