# Peer review of "A Novel Biplex Onchocerca volvulus Rapid Diagnostic Test Evaluated Among 3- to 9-Year-Old Children in Maridi, South Sudan"

_diagnostics, 2025, doi:10.3390/diagnostics15050563_

Round 1
Reviewer 1 Report
Comments and Suggestions for Authors
The study by Amber Hadermann et al., devoted to the comparative characteristics of two LFIA tests for serodiagnosis of Onchocerca volvulus, reveals the advantages of using multiantigen compositions compared to a monoantigen test. This study is of interest to readers and can be published in the journal "Diagnostics" with a little clarification.
The authors should explain the reason for choosing the IgG4 subclass as an analyte while ignoring other antibody subclasses.
The authors point out that: "Ov16 is expressed by infective L3-stage larvae, and therefore humans can develop antibodies to Ov16 when exposed to an infective bite, even if the infective L3 larvae do not mature into adults [9]. In contrast, Ov33.3 is expressed by adult worms, and OvOC3261 and OvOC10469 are expressed by the microfilariae..."
What is the rationale for the distribution of antigens in test lines 1 (Ov16 and OvOC3261) and 2 (Ov33.3 and OvOC10469)?
Author Response
Response to reviewer
We thank the reviewer for its constructive comments
We here responded point-by-point to these comments
REVIEWER 1
Comment 1
The study by Amber Hadermann et al., devoted to the comparative characteristics of two LFIA tests for serodiagnosis of Onchocerca volvulus, reveals the advantages of using multiantigen compositions compared to a monoantigen test. This study is of interest to readers and can be published in the journal "Diagnostics" with a little clarification.
The authors should explain the reason for choosing the IgG4 subclass as an analyte while ignoring other antibody subclasses.
Response 1
The IgG4 subclass was used as this was the type of IgG that was induced by the commercially available Ov16 antigen test. IgG4s are typically overexpressed for helminth infections and lead to higher specificity values than total IgG. We have added to the introduction:
“The IgG4 subclass was chosen because it is the primary immunoglobulin isotype associated with chronic helminth infections such as O. volvulus [1]. Although IgG4 represents only a minor fraction of total IgG, it is believed to be produced in response to prolonged antigenic stimulation and correlates closely with infection intensity. In a non-human primate model, IgG4 responses were closely linked to microfiladermia and declined in parallel with parasite clearance, whereas other antibody subclasses (e.g. IgM, IgG2, and IgG3) were less informative [2].”
Comment 2
The authors point out that: "Ov16 is expressed by infective L3-stage larvae, and therefore humans can develop antibodies to Ov16 when exposed to an infective bite, even if the infective L3 larvae do not mature into adults [9]. In contrast, Ov33.3 is expressed by adult worms, and OvOC3261 and OvOC10469 are expressed by the microfilariae..."
What is the rationale for the distribution of antigens in test lines 1 (Ov16 and OvOC3261) and 2 (Ov33.3 and OvOC10469)?
Response 2
The antigens were distributed in such a manner because, based on published ELISA results, this is the combination that would have led to the best combination of sensitivity and specificity [3].
REVIEWER 2
Comment 1
Your manuscript “A novel biplex Onchocerca volvulus rapid diagnostic test: Feasibility and comparison with the Ov16 SD Bioline rapid diagnostic test among 3–9-year-old children in Maridi, South Sudan” describes the evaluation of novel “biplex A” rapid diagnostic test (RDT) for Onchocerca volvulus.
The manuscript gives an interesting information; however in current form it could not be published since needs serious revision.
First, I suppose to shorten the title of the manuscript; in current form, it is too large and overloaded.
Response 1
We prefer to keep the title as it is because it explains the type of study that was conducted.
An alternative would be the following shorter title “A novel biplex Onchocerca volvulus rapid diagnostic test evaluated among 3–9-year-old children in Maridi, South Sudan”
We leave it up to the editor to make a final decision.
Comment 2
Second, the correction of English is required throughout the text. Moreover, a serious revision of the text is required. Jargon should be avoided, and the text should be carefully proofread and corrected stylistically.
Response 2
We have revised the manuscript and corrected the English. We also rewrote the text to increase its readability.
Comment 3
General remark:
If you say that the goal of research was not to evaluate the specificity of the test developed (line 107-108), how can you evaluate and discuss the seropositivity rates of the people you studied if there is a possibility that the test is not specific? It turns out that these estimates cannot be reliable.
Response 3
The specificity has already been evaluated by CDC using sera from other infections and found the specificity to be >99%. The point is that it is impossible to determine the specificity of an antibody assay in an endemic setting. Antibodies to Ov16, OvOC3261, OvOC10469, and Ov33.3 remain in circulation for decades after an infection has cleared. It is therefore not possible to say with certainty that because a person is not actively infected at the time of testing, they have no history of prior infection.
We now include in the discussion: “The specificity of the DDTD RDT prototype has already been evaluated by CDC using sera from other infections and found the specificity to be >99%. However, it is impossible to determine the specificity of an antibody assay in an endemic setting. Antibodies to Ov16, OvOC3261, OvOC10469, and Ov33.3 remain in circulation for decades after an infection has cleared. It is therefore not possible to say with certainty that because a person is not actively infected at the time of testing, they have no history of prior infection.“
Comment 4
Some comments:
L17-18 – “for confirming Onchocerca volvulus transmission” – not transmission, but infection. Correct, please, throughout the text.
Response 4
The Ov16 SD Bioline rapid diagnostic test measures the presence of O. volvulus antibodies but not the presence of infection. The detection of Ov16 antibodies in 3-9 children means that in the last 3-9 years there has been O. volvulus transmission.
We now include in the introduction
“The presence of Ov16 antibodies in young childrenthere is a mesure of infection and therefore suggests ongoing O. volvulus transmission.“
Comment 5
L23-24 – “at test line 1… at test line 2” - without a description of the detection method, such details are confusing, I suppose to delete this here. The same thing with “Ov16 and OvOC3261… Ov33.3 and OvOC10469” – what does it mean?
Response 5
Both the DDTD RDT prototype and the Ov16 RDT are lateral flow assays (LFA). LFAs operate on the principle of capillary action, where a blood sample moves across a test strip composed of one or two test lines and one control line. The latter is used to ensure the test is working properly by binding excess conjugate, regardless of the presence of the target. A test line contains immobilized antigens that bind to antibodies if they are present in the blood, forming a visible line if the antibodies are present. In the DDTD RDT Test, Line 1 contains the O. volvulus antigens Ov16 and OvOC3261 Test line 2 contains the O. volvulus antigens Ov33.3 and OvOC10469.
Ov in the codes refers to Onchocerca volvulus
OC refers to onchocerca
The numbers correspond to a unique number in an antigen bank
These Ag codes are frequently used in the onchocerciasis literature. Therefore, we do not think there is a need to explain these codes in the text.
Comment 6
L27 – “anti-Ov16 seroprevalence” – what is Ov16? Should be explained or deleted from the abstract.
Response 6
OV16 is an antigen of the O. volvulus worm which elicits strong IgG4 responses in infected individuals. For this reason, this antigen is used for serological tests. The Ov16 SD Bioline rapid diagnostic test detects IgG4 antibodies against the O. volvulus OV16 antigen, indicating past or present infection.
Comment 7
L1-32 – “Both lines were simultaneously VISIBLE IN 15.5% (37/239) OF CHILDREN” – sound confusing.
Response 7
In 15.5% (37/239) of children both the lines were visible: this means there was the presence of antibodies in the blood of the children against both Line 1 O. volvulus antigens Ov16 and OvOC3261 and Line 2 O. volvulus antigens Ov33.3 and OvOC10469.
Comment 8
L43 – “Ov16 antibody seroprevalence” – what is Ov16? Should be explained first.
Response 8
OV16 is an antigen of the O. volvulus worm which elicits strong IgG4 responses in infected individuals. For this reason, this antigen is used for serological tests. We now include this in the introduction.
Comment 9
In general, there is a lack of description of the parasite itself and which of its proteins are used for its detection by the described methods (this should be provided first, before you describe the test systems currently used).
Response 9:
We now expanded the introduction with additional information about onchocerciasis and parasite biology.
We now include in the introduction “Onchocerciasis or river blindness, caused by infection with Onchocerca volvulus is a neglected tropical disease that is associated with significant morbidity and disability including skin and eye lesions [4] and onchocerciasis-associated epilepsy [5]. The life cycle of O. volvulus involves both a human host and a blackfly (Simulium species) vector. Infective third-stage larvae (L3) are transmitted to humans through the bite of an infected blackfly. The larvae mature into adult worms in subcutaneous nodules, where they can live for years and produce the microfilariae—the primary agents of disease pathology [4].
The antigens of O. volvulus include a large variety of structural, metabolic, and excretory-secretory proteins that play crucial roles in the parasite's survival, immune evasion, and host-pathogen interactions. These antigens are also key targets for diagnostics, vaccines, and drug development. A total of 7,774 O. volvulus proteins were identified across all of the stages [6]. The OV16 antigens have been shown to elicit strong IgA responses and are therefore used as the main antigen in serodiagnostic tests. Other antigens such as the OvOC3261, Ov33.3 and OvOC10469 antigens are currently under evaluation [3]. “
Comment 10
L53 – “Ov16 SD Bioline RDT” – the manufacturer should be stated.
Response 10
We added the manufacturer: Abbot Diagnostics Inc, Giheung-gu, Yongin-si, Gyeonggi-do, 17099, Republic of Korea
Comment 11
Section 2.1 – The map is needed.
Response 11
We now included a Figure
Comment 12
L153 – “two-hour hour session” - should be corrected.
L199 – “. DDTD biplex A” – correct, please.
L231 – “. Effect of timing” – correct, please.
Response 12
Corrected
Comment 13
Figure 1 – What do the pieces of paper at the bottom with the times signed and crossed out mean? Are they needed here? In my opinion, they only confuse the reader.
Response 13
The pieces of paper contain the time of testing, the times the test had to be read and the code of the study participant. We now include this information in the legend of the figure.
Comment 14
References section - The references do not have the DOI, which would make it much easier for the reader to reference. Add, please.
Response 14
DOIs were added
We akso added the following references
- Weil, G.J.; Ogunrinade, A.F.; Chandrashekar, R.; Kale, O.O. IgG4 subclass antibody serology for onchocerciasis. J Infect Dis 1990, 161, 549-554, doi:10.1093/infdis/161.3.549.
- Cama, V.A.; McDonald, C.; Arcury-Quandt, A.; Eberhard, M.; Jenks, M.H.; Smith, J.; Feleke, S.M.; Abanyie, F.; Thomson, L.; Wiegand, R.E., et al. Evaluation of an OV-16 IgG4 Enzyme-Linked Immunosorbent Assay in Humans and Its Application to Determine the Dynamics of Antibody Responses in a Non-Human Primate Model of Onchocerca volvulus Infection. Am J Trop Med Hyg 2018, 99, 1041-1048, doi:10.4269/ajtmh.18-0132.
- Bennuru, S.; Oduro-Boateng, G.; Osigwe, C.; Del Valle, P.; Golden, A.; Ogawa, G.M.; Cama, V.; Lustigman, S.; Nutman, T.B. Integrating Multiple Biomarkers to Increase Sensitivity for the Detection of Onchocerca volvulus Infection. J Infect Dis 2020, 221, 1805-1815, doi:10.1093/infdis/jiz307.
- Haffner, A.; Guilavogui, A.Z.; Tischendorf, F.W.; Brattig, N.W. Onchocerca volvulus: microfilariae secrete elastinolytic and males nonelastinolytic matrix-degrading serine and metalloproteases. Exp Parasitol 1998, 90, 26-33, doi:10.1006/expr.1998.4313.
- Hadermann, A.; Amaral, L.J.; Van Cutsem, G.; Siewe Fodjo, J.N.; Colebunders, R. Onchocerciasis-associated epilepsy: an update and future perspectives. Trends Parasitol 2023, 39, 126-138, doi:10.1016/j.pt.2022.11.010.
- Bennuru, S.; Cotton, J.A.; Ribeiro, J.M.; Grote, A.; Harsha, B.; Holroyd, N.; Mhashilkar, A.; Molina, D.M.; Randall, A.Z.; Shandling, A.D., et al. Stage-Specific Transcriptome and Proteome Analyses of the Filarial Parasite Onchocerca volvulus and Its Wolbachia Endosymbiont. mBio 2016, 7, doi:10.1128/mBio.02028-16.
Reviewer 2 Report
Comments and Suggestions for Authors
Dear authors,
Your manuscript “A novel biplex Onchocerca volvulus rapid diagnostic test: Feasibility and comparison with the Ov16 SD Bioline rapid diagnostic test among 3–9-year-old children in Maridi, South Sudan” describes the evaluation of novel “biplex A” rapid diagnostic test (RDT) for Onchocerca volvulus.
The manuscript gives an interesting information; however in current form it could not be published since needs serious revision.
First, I suppose to shorten the title of the manuscript; in current form, it is too large and overloaded.
Second, the correction of English is required throughout the text. Moreover, a serious revision of the text is required. Jargon should be avoided, and the text should be carefully proofread and corrected stylistically.
General remark:
If you say that the goal of research was not to evaluate the specificity of the test developed (line 107-108), how can you evaluate and discuss the seropositivity rates of the people you studied if there is a possibility that the test is not specific? It turns out that these estimates cannot be reliable.
Some comments:
L17-18 – “for confirming Onchocerca volvulus transmission” – not transmission, but infection. Correct, please, throughout the text.
L23-24 – “at test line 1… at test line 2” - without a description of the detection method, such details are confusing, I suppose to delete this here. The same thing with “Ov16 and OvOC3261… Ov33.3 and OvOC10469” – what does it mean?
L27 – “anti-Ov16 seroprevalence” – what is Ov16? Should be explained or deleted from the abstract.
L31-32 – “Both lines were simultaneously VISIBLE IN 15.5% (37/239) OF CHILDREN” – sound confusing.
L43 – “Ov16 antibody seroprevalence” – what is Ov16? Should be explained first.
In general, there is a lack of description of the parasite itself and which of its proteins are used for its detection by the described methods (this should be provided first, before you describe the test systems currently used).
L53 – “Ov16 SD Bioline RDT” – the manufacturer should be stated.
Section 2.1 – The map is needed.
L153 – “two-hour hour session” - should be corrected.
L199 – “. DDTD biplex A” – correct, please.
L231 – “. Effect of timing” – correct, please.
Figure 1 – What do the pieces of paper at the bottom with the times signed and crossed out mean? Are they needed here? In my opinion, they only confuse the reader.
References section - The references do not have the DOI, which would make it much easier for the reader to reference. Add, please.
Comments on the Quality of English LanguageThe correction of English is required throughout the text
Author Response

(The authors gave the same response as above.)

Round 2
Reviewer 2 Report
Comments and Suggestions for Authors
Dear authors,
Thank you for your reply and the revision of the manuscript.
Thank you for the changes made according to my suggestions and comments. You did a lot of work, and it became truly easier to understand.
I still insist that the current title of the manuscript is much overweighed. The variant you suggested looks better, if possible, it would be better to replace the title with new one.
Some minor comments to solve:
L59 – “Simulium” – should be in italic.
L86-87 – “(Abbot Diagnostics Inc, Gi-86 heung-gu, Yongin-si, Gyeonggi-do, 17099, Republic of Korea)” – no need to provide the full address of the manufacturer, “(Abbot Diagnostics Inc, Republic of Korea)” will be enough.
Figure 1 – the legend “Scale 1:1” is confusing. In addition, the map scale should be indicated - are distances shown in kilometers? What do the black and red circles mean? If this drawing is taken from somewhere, you should provide the source/reference and scan it in better quality.
L134 – “…accessed on” – please add the date.
Author Response
Response to reviewer
I still insist that the current title of the manuscript is much overweighed. The variant you suggested looks better, if possible, it would be better to replace the title with new one.
Response
We now changed the title to “A novel biplex Onchocerca volvulus rapid diagnostic test evaluated among 3–9-year-old children in Maridi, South Sudan”
Some minor comments to solve:
L59 – “Simulium” – should be in italic.
Response
Done
L86-87 – “(Abbot Diagnostics Inc, Gi-86 heung-gu, Yongin-si, Gyeonggi-do, 17099, Republic of Korea)” – no need to provide the full address of the manufacturer, “(Abbot Diagnostics Inc, Republic of Korea)” will be enough.
Response
We dropped the full address
We now only mention Abbot Diagnostics Inc, Republic of Korea
Figure 1 – the legend “Scale 1:1” is confusing. In addition, the map scale should be indicated - are distances shown in kilometers? What do the black and red circles mean? If this drawing is taken from somewhere, you should provide the source/reference and scan it in better quality.
Response
The figure is a drawing made by our research team in South Sudan. The figure was published previously and we cite the reference. We uploaded a new cleaner Figure.
L134 – “…accessed on” – please add the date.
Response
We added the dates
